# Nectar Dynamics and Pollinators Preference in Sunflower

**DOI:** 10.3390/insects13080717

**Published:** 2022-08-09

**Authors:** Simone Bergonzoli, Elio Romano, Claudio Beni, Francesco Latterini, Roberto Lo Scalzo, Antonio Scarfone

**Affiliations:** 1Consiglio per la Ricerca in Agricoltura e l’Analisi dell’Economia Agraria (CREA), Centro di Ricerca Ingegneria e Trasformazioni Agroalimentari, Via della Pascolare 15, 00015 Monterotondo, Italy; 2Consiglio per la Ricerca in Agricoltura e l’Analisi dell’Economia Agraria (CREA), Centro di Ricerca Ingegneria e Trasformazioni Agroalimentari, Via Milano 43, 24047 Treviglio, Italy; 3Institute of Dendrology, Polish Academy of Sciences, Parkowa 5, 62-035 Kórnik, Poland; 4Consiglio per la Ricerca in Agricoltura e l’Analisi dell’Economia Agraria (CREA), Centro di Ricerca Ingegneria e Trasformazioni Agroalimentari, Via Giacomo Venezian 26, 20133 Milano, Italy

**Keywords:** compost, soil fertilization, insect habits, hybrid, non-hybrid varieties

## Abstract

**Simple Summary:**

The survival of pollinators is in real danger, and the reasons are due to climate change, inadequate pesticide use, invasive species and diseases. The nectar produced by plants is the basic food for many of these insects, together with pollen. However, the scarcity of nectar secretion in certain plant species is becoming alarming, especially in recent years. We focused our study on sunflower species, a plant which is an object of debate within the scientific community for the strong industrial interest, but also for the decline in pollen, nectar and honey production displayed globally in recent decades. We tested a commercial hybrid variety in different soil conditions and verified nectar secretion and quality. We also evaluated the pollinator visiting habits using the same hybrid in comparison with a non-hybrid variety. Our findings point out the effect of compost with respect to nectar composition and the pollinators preference toward the non-hybrid variety.

**Abstract:**

Nectar is a complex biochemical substance secreted with particular rhythm by flower nectaries. Nectar is the base of a mutualism in which pollinators consume nectar, as food source, and are involuntarily responsible for the transport of pollen and pollination. The dynamics and temporal patterns of nectar secretion are still not fully understood as well as the environmental and climatic factors influencing its production. The quantity and quality of nectar found in standing crops at flowering influence the mutualistic relationship with pollinators and their foraging behavior. This situation is even more significant considering the reduction in undisturbed environments, the loss of soil quality, the spread of monoculture agricultural management and the use of self-fertile hybrids. The objects of the study are understanding the relationship among soil properties and nectar quality, comparing the nectar composition in a sunflower hybrid variety and evaluate pollinator preferences in selecting nectar sources among hybrid and non-hybrid varieties. For these purposes, two different experimental tests were established. Results highlighted that fertilization strategy influenced crop biomass development, determined soil characteristics and nectar composition in Sunflower. However, when comparing nectar composition of hybrid and non-hybrid varieties of sunflower, no significant differences were found. Despite this, the analysis of number of visits on the two treatments showed statistically significant differences. This research provides further understanding of the very complex relationship among soil, crop and nectar to support the definition of agricultural management strategies and reach the optimal nectar composition level for pollinators in agricultural crops.

## 1. Introduction

Sunflower (*Helianthus annuus* L.) is a herbaceous cross-pollinated plant that belongs to the family of Asteraceae, requiring insects to achieve pollination for seed production [1]. The species is considered one of the most important crops for oilseed production, with a global cultivated surface of 27 Mha and an estimated seed production of 52 Mtons per year [2,3]. In Italy, the cultivation of sunflower is concentrated in the central part of the peninsula [4], where climatic conditions allow rainfed cultivations covering a total surface of 123,000 ha [5]. Sunflower is also considered an important species for honeybee production, being ranked among the principal plants for melliferous potential [6]. Sunflower honey is also highly demanded for its aromatic and nutritional properties, which includes the abundance of nutraceutical and antioxidant substances as polyphenolic compounds [7]. Despite the large interest for this species in agriculture, beekeeping and food markets, studies suggest that in recent decades, the honey production from sunflower has decreased dramatically worldwide [8,9,10].

Many theories have been put forward to explain for this decline, and global warming due to climatic change is undoubtedly considered one of main factors affecting the nectar production, as nectar secretion relies much on optimal environmental factors, especially in rainfed herbaceous crops such as sunflower [11,12,13,14,15]. Indeed, studies suggest that under both experimental and natural conditions, the nectar volumes and the nectar production rates in sunflower decrease at higher temperatures, high plant water stress (influenced by soil moisture), and low relative air humidity [6]. Nectar is an important floral resource playing a fundamental role as an energetic resource for pollinators [16]. Apart from its involvement in a mutualistic relationship with pollinators [17], nectar production dynamics are not completely understood and the variability in its quality and production may be related to environmental, ecological and morpho-anatomical factors. Recent studies have suggested that this variability can also manipulate pollinator behavior during or after visits, which can directly affect pollen transfer and plant reproduction [18,19]. This was observed also in pollen, where the specialization of certain insects play a key role for the survival of certain species [20].

However, despite global warming and climate change, other causes generally attributed to the decline of sunflower honey production include the availability of the pest of the insect, the pesticide exposure, the beekeeping mismanagement, the long-distance transport, and the decreased genetic diversity [13]. In particular, the introduction of new self-fertile hybrids respect to old population seems another verified option [8,9]. In a study performed in India, six hybrids and two old sunflower cultivars were tested with respect to nectar secretion and honey production capacity, utilizing a quantification method based on dry nectar sugar (DNS). The researchers attributed the missing honey production to the scarce production of nectar in the majority of the new hybrids (83.3%), while the old cultivars far exceeded the hybrids in all DNS parameters evaluated [9]. However, there are still many doubts regarding the nectar production decline in sunflower and the existing potential relationship between soil properties, plant cultivars and pollinators. Studies on different plant species revealed different results according to the soil characteristics analyzed [21,22,23,24,25].

In a review by David et al. (2019) [21], it was investigated if changing in soil nitrogen could affect plant–pollinator interactions. The authors concluded that floral traits relevant to pollinators, such as phenology, morphology, nectar production, pollen production and quality can be affected by soil N, but a lack of knowledge was still identified to determine if and how pollinators will be impacted by these changes. In a study performed in New Zealand [22], the influence of soil chemical properties was investigated on three mānuka (*Leptospermum scoparium* J.R.) cultivars utilizing soils from different locations and evaluating the plant growth, the flowering phenology, and the nectar production and quality. The study suggests that higher nectar yield in certain cultivars was related to soil moisture stress. In addition, greater plant growth and floral density was achieved in response to specific soil nutrients, suggesting the potential to improve nectar yield by using targeted fertilization. Other studies suggest that soil quality enhancement achieved with vermicompost amendment can significantly affect plant-pollinator interactions and directly influences pollinator nutrition and overall performance [21]. On the other hand, the soil nutrient enrichments in some cases determined problems to pollinators. For instance, in a study conducted by Ceulemans et al. (2017) [26], the addition of soil nutrients as fertilizer in *Succisa pratensis* Moench species determined the alteration of amino acids and sugar composition of plant nectar and pollen, increasing the larval mortality of its natural pollinator *Bombus terrestris* L. Therefore, the relationship among soil properties and nectar composition is not completely understood. In the present study, the specific relationship mentioned above has been investigated testing a sunflower “high oleic” hybrid variety in a field experiment using different soil treatments. A further goal of this study was to compare the nectar composition in two sunflower varieties (one hybrid and one non-hybrid) grown in the same conditions. The last objective was to quantify the visiting habits of pollinators on two sunflower varieties (one hybrid and one non-hybrid). For these purposes, three different experimental tests were established: a study on the influence of soil treatments on nectar composition, a specific study on nectar composition of hybrid and non-hybrid varieties, and a pollinators study.

## 2. Materials and Methods

### 2.1. Experimental Area

The experimental field was set up at CREA-IT of Treviglio (Bergamo, Italy) (45°31′17.18′′ N; 09°33′50.82′′ E) during the period April–September 2021. On a field of about one hectare, which was not cultivated before the test, the soil was ploughed and harrowed to sow sunflower (*Helianthus annuus* L.) on 7th April 2021, selecting a hybrid variety (Ref. N° LST 907). Sowing was performed in order to reach a density of 7 plants per m^2^. The site of the experiment was identified within the field by selecting a homogeneous area according to the soil map obtained via automatic resistivity profiling (ARP). The main physical and chemical characteristics of the soil before starting the test were studied by collecting three soil samples randomly selected from the study area. Crop was grown under rainfed conditions.

### 2.2. Soil Treatments Influence on Nectar Composition

In order to study the influence of soil on nectar, three soil treatments were identified: CONTROL (no fertilizer or compost application), CHEMICAL (only mineral fertilizer application), COMPOST (only compost application).

Nine plots, three per treatment, of 16 m^2^ (4 m × 4 m) were organized in three blocks (three plots per block) according to a randomized block experimental design.

Regarding the treatment CHEMICAL (CHEM), after the sowing, 30 g m^−2^ of chemical fertilizer was applied for a total of 480 g per plot corresponding to 300 kg ha^−1^. The same fertilization was also applied on 1st of June at flower bud development stage. The fertilizer applied was NPK 15-15-15.

Concerning the treatment COMPOST (COM), before harrowing, 3 kg m^−2^ of compost was applied for a total of 48 kg per plot corresponding to 3 Mg ha^−1^. No further application of compost was applied. The compost used was characterized as follows: 50% moisture content, 8 pH, salinity 1.5 dS m^−1^, organic C 20%, and C/N ratio 25.

For treatment CONTROL (CON), no fertilization was applied during all the crop development.

### 2.3. Crop Development

During crop development, the physiological status of plants was evaluated using Dualex (ForceA, Université Paris Sud-Bât 503, Rue du Belvédère, 91893 ORSAY CEDEX France), by a leaf clip sensor to measure chlorophyll and polyphenols indexes of plant leaves. This optical sensor allows non-destructive estimation of chlorophyll, flavonols and anthocyanins in leaves and calculate the NBI^®^ (Nitrogen Balance Index), which combines chlorophyll and flavonols contributions (related to nitrogen/carbon allocation). Sampling was performed by testing upper and lower side of the highest leaf of the plant, for 12 different plants of each plot, and was carried out on 22nd June and 7th July. The instrument was calibrated following its specific procedure before starting to test each treatment.

### 2.4. Nectar Sampling

In order to measure nectar composition and concentration of sugars, the washing method was used [27] on flowers collected during sampling on 28 June and on 5 July. As for the quantification of sugars, the methodology proposed by Rinku and Chaudhary (2017) [28] was followed, with some modifications. Specifically, ten flowers at the same stage were collected from the same inflorescence and sealed in a Falcon test tube with 5 mL of distilled water; the weight of the ten flowers was 0.1 g on average. Three samples for each plot were collected for a total of 27 samples, the inflorescence of which was collected, and the flowers were covered with a plastic tulle mesh to prevent nectar consumption from pollinators. Samples were directly transported to the lab to be analyzed. Analysis performed regarding the extract of the flowers treated with water, vortexed for 20 s, and ultrasounded for 5 min, then further subjected to filtration on glass wool.

The level of soluble solids residue (SSR) was measured on the aqueous phase by refractometry, using a Bellingham-Stanley RFM 91 multi-scale digital apparatus, and unity of measure used for SSR was Brix (°Bx), taken from the dried units of SSR for a single flower (°Bx/flower).

Single sugars concentrations were measured by HPLC, injecting an aliquot of the clean aqueous flower extract into the chromatograph (Jasco system), equipped with a column specific for carbohydrate separation, Benson polymeric, Ca++, 300 per 8.7 mm, kept at 80 °C, with mobile phase water at a flow 0.7 mL/min. In these conditions, the retention times of the identified and quantified simple sugars were 6.0 min (polysaccharides at low molecular weight, composed of 5 or 4 sugars units, DP5 and DP4), 7.2 (raffinose), 8.4 (sucrose), 10.1 (glucose), 12.1 (fructose) and 14.9 (mannitol). The data analysis from the output of the chromatographic system were managed by a Clarity software, version 2.6.5.517, DataApex Ltd., Prague, Czech Republic. The calibration of the chromatographic system was achieved by using solutions at known concentrations of pure compounds (nystose for DP4 and DP5 sugars), and the quantitative data were obtained by interpolation of peak areas from the flower extract with calibration curves (r^2^ = 0.993–0.999). Quantitative data were given as dried nectar secretion, in μg/flower, as dried sugars content, following the methodology established by Rinku and Chaudhari (2017) [28].

### 2.5. Biomass Collection

At day 1st of August the total biomass of each plot was harvested, collecting the aboveground biomass. Stems with leaves and flowers were weighted separately using the scale RADWAG WLC6/C1/R, Radom, Poland with 0.1 g sensitivity. Dry weight and moisture content of each part of the biomass were estimated according to EN ISO 18134-2:2017 standard [29].

### 2.6. Soil Sampling

After biomass harvesting, soil samples were collected from each plot in order to study soil quality variation. One sample of 500 g from each plot was collected and shipped to the lab in order to be analyzed. Each sample was collected following the soil sampling methodology (ISO 18400-205:2018) [30]. Parameters studied were: Clay, Silt, Sand, pH, total organic C, total N, available P (Olsen), exchangeable K.

### 2.7. Comparison of Nectar Composition

A second field experiment was set up in a plastic greenhouse of 40.5 m^2^ (9 m × 4.5 m) to compare nectar composition of a hybrid variety and a non-hybrid variety. Inside the greenhouse, the soil was sown with two varieties of sunflower (*Helianthus annuus* L.), the left part sown with a non-hybrid variety “Irish eye” and the right part with the hybrid variety (Ref. N° LST 907), utilized in the other field experiment. Front and back part of the greenhouse were opened during the test to allow ventilation and to reduce the inner temperature to allow pollinators activity. Sowing was performed manually on 7th April 2021 with a plant density of 7 plants per m^2^, with the same quantity as that used in other field experiment. No fertilization was applied while irrigation was applied once per week from 15th of April to 15th July. Nectar sampling was performed in the same dates of the other experiment and following the same methodology, as described in Section 2.4, collecting three samples per treatment for a total of six samples.

### 2.8. Pollinator Study

In the same plastic greenhouse described in Section 2.8, a further experiment was set up to study the visiting habits and preferences of some pollinator groups, namely honeybees (*Apis mellifera*) and *Bombus* spp. insects. In the experiment, 2 cameras were installed for each variety, for a total of 4, capable of acquiring images of one sunflower inflorescence per camera that was selected for similar height and area. The images were acquired from a single-board computer with wireless LAN and Bluetooth connectivity Raspberry pi3 Model B + (1.4 GHz 64-bit quad-core processor, dual-band wireless LAN) and a v2. The camera has an 8 megapixel Sony IMX219 sensor and has been programmed using the Picamera Python library program. The images were acquired every minute, lasting 1.5 h, from 10:00 am to 11:30 am, for 7 consecutive days starting from the flowering stage.

Subsequently, an analysis methodology was applied to the images for counting the pollinating insects present in each inflorescence of each image using the binarization of the images based on the thresholds that discriminated the colors of the bees with respect to the corollas on which they were laying.

### 2.9. Data Analysis

Statistical analyses of data of soil and leaf sampling, biomass harvesting, and nectar characteristics were conducted with the Comprehensive R (R Core Team 2021) Archive Network (CRAN) software (Institute for Statistics and Mathematics, Wien-Umgebung, Austria) [31].

The “MissMech” Package was used for testing homoscedasticity and normality, after checking data, and the analysis of variance was developed through the “stats” package and the ANOVA test for the verification of the statistical significance of the effect of the observed independent variables (see Tables, letter of significance) and the consequent post hoc tests (i.e., Tukey) [32] for the evaluation of the difference between the variants. This procedure was used for all the parameters studied in the first experiment that showed a *p*-value < 0.05 after ANOVA test. For the analysis of pollinator visit regarding the varietal study, the *t*-test was used for the verification of the statistically significant differences among treatments.

## 3. Results

### 3.1. Soil Treatments

Results of soil sampling regarding the different treatments are depicted in Table 1.

The results of soil sampling highlighted that the soil under study was a medium loam soil (USDA). The results of clay, silt and sand confirmed the homogeneity of the area of experiment concerning physical characteristics. All the treatments showed an increase in total organic C, N, P and K with respect to the sampling before ploughing. Considering values of clay, sand, silt and pH, no statistically significant differences were found among treatments. Regarding the results of organic C, treatment COMP and CHEM showed higher values (3 and 2.7%, respectively) with respect to other treatments (2.3 and 2.2% for CON and before ploughing, respectively) with statistically significant differences. Concerning N, treatment COMP highlighted higher values (0.24%) with respect to all other treatments with statistically significant differences. Considering the results of P, the treatments COMP, CHEM and CON showed higher values (226, 227 and 211 mg/kg d.m., respectively), with statistically significant differences, respect to the “Before ploughing” treatment (149 mg/kg d.m.). The same situation was found considering results of K; treatments COMP, CHEM and CON showed higher values (306, 224 and 252 mg/kg d.m., respectively), with statistically significant differences with respect to the “Before ploughing” treatment (204 mg/kg d.m.). The results highlighted that concerning the chemical parameters, the experimental test caused an increase in the values with respect to the situation before ploughing. Concerning P and K, no differences were found among CON, CHEM and COMP treatments. Only the treatment COMP showed statistically significant differences for each chemical parameter with respect to the “Before ploughing” condition. The results of C, P and K of COMP and CHEM did not show statistically significant differences.

### 3.2. Crop Development and Biomass Collection

In Table 2, the results of the two leaf samplings by using the Dualex instrument are described. The values of NBI^®^ index are expressed as a range between 0 and 100, with the values of chlorophyll as a range between 0 and 150 and the values of flavonols and anthocyanins as a range between 0 and 3.

The results of leaf sampling highlighted very similar values between treatments and among the two sampling dates for each parameter. No statistically significant differences were found. Regarding the first sampling, treatment COMP resulted in a higher value of chlorophyll (38.22), a lower value of flavonols (1.17), with a consequent higher value of NBI^®^ (43.74). Concerning the second sampling treatment, COMP showed slightly a lower value of flavonols (1.28) and a higher value of NBI^®^ (37.99). Considering anthocyanins, the values were very low with respect to the range of the parameter (0 to 3), and they were very similar among treatments (0.16) and slightly higher in the second sampling (0.18).

The results of biomass collection (Table 3) showed very similar results for stem weight of treatments CHEM and COMP (24.57 and 24.97 kg f.m., respectively) but higher than CON (19.97 kg f.m.), with statistically significant differences among treatments. Regarding the stem moisture content, statistically significant differences were found among treatment COMP (79.12%) and treatments CHEM and CON (76.83 and 75.68%, respectively). Treatment COMP showed a higher moisture content, but it showed the same amount of stem biomass collected. No significant differences were found regarding inflorescence weight and its moisture content.

### 3.3. Nectar Characteristics

In Table 4, the results of flower extracts composition collected from different treatments of the plots are described.

Considering the first sampling, the total sugars concentration of the exudates showed statistically significant differences between treatments CHEM and COMP (average of 445 and 345 mg/flower, respectively) and treatment CON, which highlighted a more concentrated exudate, with 670 mg/flower (Table 4).

The components more relevant to the flower extract were glucose and fructose for each treatment, representing 40% and 47% of the total for CON, 37% and 51% for CHEM, and 30% and 50% for COMP, respectively, with fructose as the main component for each treatment, highlighting a high internal variability, especially in the first sampling (see data of std deviation, Table 4). However, statistically significant differences were found between treatments CHEM and COMP (average of 230 and 175 mg/flower, respectively) and treatment CON (320 mg/flower), in a close relationship with the case of total sugars concentration (Table 4). Results of °Brix were very similar between the treatments without statistically significant differences. The ratio Total/Brix showed higher values of CON (0.79) with respect to CHEM (0.59) and COMP (0.49), with statistically significant differences. Considering the second sampling, the total concentration of the flower extracts showed statistically significant differences among treatments, with COMP highlighting more diluted exudate, with a lower average sugar content (337 mg/flower) than CON and CHEM (Table 4), confirming the data from the first sampling.

For the second sampling, the components more relevant to the exudate were also glucose and fructose for each treatment, representing 31% and 57% of the total for the CON, 31% and 54% for CHEM and 31% and 53% for COMP, respectively. Fructose was also the main component for each treatment for the second sampling date, with a minimal internal variability if compared with the previous sampling (see std deviation data, Table 4). Statistically significant differences were found between treatments CON (270 mg/flower), CHEM (250 mg/flower) and treatment COMP (180 mg/flower), fully confirming the case of total concentration. The results of °Brix were very similar between the treatments without statistically significant differences. The ratio Total/°Brix showed only a tendency to higher values of CON and CHEM with respect to COMP, with no statistically significant differences, resembling the data from the first sampling.

### 3.4. Pollinator Study

In Table 5, the results of nectar composition collected from treatments of the greenhouse, comparing two different sunflower genotypes sampled in two distinct periods, are reported.

Concerning the first sampling, the components more relevant to the flower extracts were glucose and fructose for each treatment, representing 34% and 43% of the total for the hybrid and 22% and 42% for non-hybrid, respectively. Fructose was the main component for each treatment, as in all previous samplings. The presence of mannitol was detected in Non-Hybrid variety in the first sampling and in both genotypes in the second sampling.

The components more relevant to the flower extract were also for the second sampling Glucose and Fructose for each treatment, representing the 27% and 52% of the total for the Hybrid and 30% and 47% for Non-Hybrid. Fructose was the main component for each treatment also for the second sampling date. The presence of mannitol was detected in the two treatments, with average values of 40 mg/flower in the non-hybrid variety of the first sampling and in both hybrid and non-hybrid varieties of the second sampling (10 and 5 mg/flower, respectively), as shown in Table 5.

Regarding the results of nectar analysis in hybrid and non-hybrid varieties, the comparison among the two samplings periods showed an increase in the values of glucose, fructose, total sugars and Brix for each treatment in the second sampling, with statistically significant differences. On the other hand, no significant differences were found between treatments. Specifically, total sugars levels changed from averages of 433 and 258 mg/flower in the first sampling to 740 and 555 mg/flower in the second sampling (Table 5) for the hybrid and non-hybrid varieties, respectively. A similar trend has been found for the average °Bx values, changing from 700 and 550 in the first sampling to 1300 and 1100 in the second sampling (Table 5) for the hybrid and non-hybrid varieties, respectively.

Glucose level increased from 150 and 60 mg/flower in the first sampling to 205 and 170 mg/flower in the second sampling (Table 5) for the hybrid and non-hybrid varieties, respectively. Similarly, the fructose level increased from 190 and 115 mg/flower in the first sampling to 390 and 265 mg/flower of the second sampling (Table 5) for the hybrid and non-hybrid varieties, respectively.

The results of the analysis of the images recorded in the greenhouse showed that only two pollinator groups were detected, namely honeybees (*Apis mellifera*) and *Bombus* spp. (species not detected). The total number of visits recorded was 506 and 1526 for hybrid and non-hybrid varieties, respectively, after 10.5 h of observations. The number of visits of the two species of pollinators increased in the first few days and then decreased, confirming that the number of visits followed the progress of flowering. After the 7th day, the number of visits was so low, and the number of flowers open in the inflorescences selected too small, that the recording was stopped. Figure 1 shows how the number of visits photographed on the flower heads of the non-hybrid variety (orange in the graph) was considerably higher than the hybrid variety (blue), although the trend was very similar. The values indicated in the graph refer to the number of visits, intended as the sum of bumblebees and honeybees, photographed during 1.5 h on two flower heads. The analysis of the number of visits photographed in the two treatments showed statistically significant differences according to the *t*-test (t = −3.8136, df = 12, *p*-value = 0.002468) performed on the number of visits photographed during 1.5 h on two flower heads of each treatment. Despite the absence of statistically significant differences among nectar of two varieties studied, a trend toward the dilution of nectar was identified that might have influenced the pollinators’ visits.

## 4. Discussion and Conclusions

The study had the objective to evaluate the effect of soil fertilization on nectar production and composition in a high oleic hybrid variety of sunflower and, in another experiment, to compare this variety with a non-hybrid one with respect to nectar production, nectar quality and pollinator preference habits. In the first part of the study, the effect of COMP and CHEM on soil characteristics was evaluated. The results showed that COMP increased all levels of chemical parameters (C, N, P and K) with respect to soil conditions before the test. Comparing the results with those of CHEM, the treatment COMP evidenced a significantly higher nitrogen content in soil, while a non-significant but increasing trend was observed for C content. As other studies suggest [33,34], the increase in C and N are indicators of the balance improvements between soil mineralization and humification, induced by both the amendment with compost and the incorporation into the soil of the residues of spontaneous vegetation prior to the field test.

Results of the study performed with dualex leaf sampling did not show significant difference among treatments. This study was performed because several studies have shown that polyphenols, specifically flavonols, are a good indicator of nitrogen status of plants. On one hand, when a plant is under optimal conditions, it favors its primary metabolism. As a result, the plant shifts its metabolism towards the proteins (nitrogen-containing molecules) containing chlorophyll, unfavoring the synthesis of flavonols (carbon-based secondary metabolites compounds). On the other hand, in the case of nitrogen deficiency, the plant directs its metabolism towards an increased production of flavonols. The NBI^®^ is less sensitive to the variations of environmental conditions than the chlorophyll (leaf age and leaf thickness among others). As a result, the higher the NBI^®^, the better the plant status. The absence of significant differences in leaf sampling parameters indicates that regarding plant status, the use of chemical fertilizer and soil amendment has no evident effects.

The study of biomass displayed a higher development of the plants treated with compost and chemical fertilizer, but it also showed that the water content was higher in stems of the COMP treatment. This can be interpreted as a positive physiological response of the plant to absorb water by using the soil amendment, which was not achieved with chemical fertilization. The increase in biomass moisture can also be linked to the higher water retention of the soil resulting from the amendment, which leads to a greater availability of water for the crop [35].

The study of nectar characterization revealed that that the use of compost in the longer term (2nd sampling period) has determined a reduction in total sugar concentration. Searching for connections between the results obtained in biomass analysis (Table 3) and those of nectar characterization (Table 4), it can be observed that the reduced concentration of total sugars is associated to the higher moisture content ofplant stem as a consequence of the soil amendment application. This finding evidences the effect of compost on sunflower nectar composition and could be further exploited in the future.

Another interesting observation derives from the Total/Brix ratio. From the results, it is evident that CHEM and COMP have a higher tendency to accumulate soluble substances other than simple sugars with respect to CON samples. For example, a significant contribution to this ratio could be given by amino acids contribution, noted for their presence in nectar [36,37]. This is in line with previous studies regarding the increase in free amino acids with increased fertilizer use. The results of the second sampling, performed in July, also highlighted the presence of mannitol for each treatment among the sugars, even in a very low concentration. This presence could possibly be related to a potential stress response, as well as osmotic balancing phenomena [38]. Interestingly, mannitol was found as a minor component of nectar from other species [39].

In the second study, the nectar composition and some pollinator habits were evaluated with respect to the sunflower hybrid already used in the first study and a non-hybrid variety. The nectar analysis displayed no difference among hybrid and non-hybrid varieties. Since plants were grown in the same conditions, with no agronomical inputs applied (irrigation, fertilization), it is possible to conclude that genetic differences due to plant variety did not affect the nectar composition of the plants. On the other hand, the number of pollinator visits on the flower heads of the non-hybrid variety was considerably higher than that observed in the hybrid variety, although the trend was similar. The literature in this regard is still uncertain. For instance, Liu et al., 2019 [40] stated that the strong linkage between soil, plants, and pollinators can influence pollination services, but the composition of nectar sources that are preferably selected by pollinators is not completely understood. A specific study of Heyneman (1983) [41] stated that because bees often travel long distances to flowers, the flight to and between flowers constitutes the most energetically costly phase of their foraging. Although bees might be expected to favor the production of even more concentrated nectars, in order to gain more energy per volume of nectar acquired, their water needs may effectively set an upper limit on acceptable sugar concentrations. Following the study of Heyeman, an observation to be noted is that despite the absence of significant differences, the total sugar content observed in the non-hybrid variety of nectar was lower, indicating a certain nectar dilution. However, this cannot represent a safe explanation of the pollinator behavior as other attractive effects may have determined the insect choices. For instance, the unknown semiochemical profilenn emitted by the different sunflower varieties differently influences the behavior of honeybees, especially in some volatile compounds emitted by sunflower [42,43]. Other factors can be the number of flowers per plant, the small sample size for pollinator observation, the flower color, the flower morphology and so on. These aspects will be considered for future studies.

In conclusion, for the sunflower hybrid analyzed, this work demonstrated that the differences in some biomass characteristics and in nectar composition can be attributed to different soil treatments. The use of compost as a soil amendment determined a better water absorption of the plants, which was highlighted by the higher moisture content of the biomass and the reduced sugar concentration in the nectar. The varietal study focused on pollinators demonstrated the strong propensity of the insects to visit the non-hybrid variety. The observed trend in nectar dilution may have influenced the insect choices, as verified in previous studies, but differences in sugar concentration of two varieties were non-significant and other factors (flower morphology, number of flowers per plant, color) may have affected the pollinator selectivity. Future studies will be conducted to further understand nectar composition dynamics and compare nectar composition in different sunflower varieties subjected to different fertilization strategies and to study the pollinator preferences regarding the selection of different sources of nectar.

## Figures and Tables

**Figure 1 insects-13-00717-f001:**
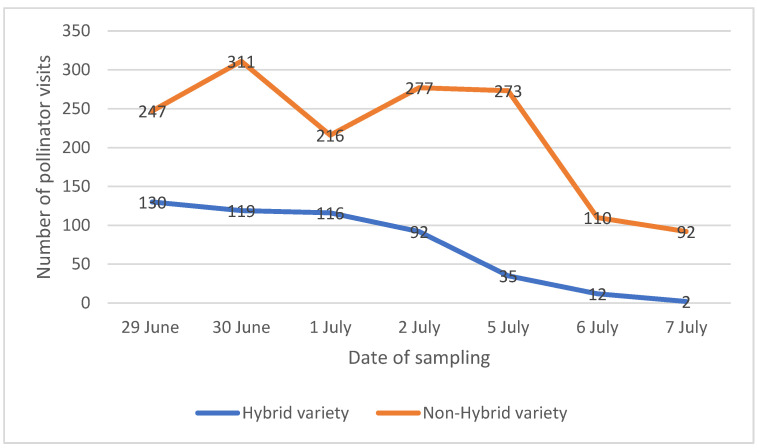
Results of image analysis of the visits of some pollinator insects on the flower heads of the two varieties. The values reported refer to the number of insects photographed during 1.5 h on two flower heads, intended as the sum of bumblebees and honeybees.

**Table 1 insects-13-00717-t001:** Main physical and chemical characteristics and standard deviation of the soil tested. In each column, values with different letters indicate mean values significantly different according to the Tukey’s post hoc test performed on results of ANOVA with *p*-value < 0.05 CON = Control; CHEM = chemical fertilizer; COMP = Compost.

Treatments	Clay (% d.m.)	Silt (% d.m.)	Sand (% d.m.)	pH	Total Organic C (% d.m.)	N (% d.m.)	Olsen P (mg/kg d.m.)	Interchangeable K (mg/kg d.m.)
Before ploughing	19.1 ± 4.3	30.5 ± 6.5	50.4 ± 10.0	7.7 ± 0.5	2.2 ± 0.4a	0.15 ± 0.02a	149.0 ± 30.0a	204.0 ± 29.0a
CON	18.7 ± 4.2	30.1 ± 6.5	51.2 ± 10.1	7.8 ± 0.5	2.3 ± 0.4a	0.18 ± 0.03a	211.0 ± 40.0b	252.0 ± 35.0b
CHEM	18.9 ± 4.2	31 ± 6.6	50.1 ± 10.0	7.8 ± 0.5	2.7 ± 0.4b	0.19 ± 0.03a	227.0 ± 42.0b	224.0 ± 31.0b
COMP	20.1 ± 4.5	29.5 ± 6.3	50.4 ± 10.0	7.7 ± 0.5	3 ± 0.5b	0.24 ± 0.04b	226.0 ± 42.0b	306.0 ± 43.0b

Clay: F 0.07, df 11, P 0.97; Silt: F 0.02, df 11, P 0.99; Sand: F 3.97, df 11, P 0.057; pH: F 1.58, df 11, P 0.26; C: F 67.6, df 11, P 5,0E-06; N: F 25.71, df 11, P 0.0001; K: F 160.8, df 11, P 1.73E-07; P: F 223.9 df 11, P 4,7E-08.

**Table 2 insects-13-00717-t002:** Results of mean value and standard deviation of leaf samplings. Values with different letters indicate mean values significantly different according to the Tukey’s post hoc test performed on results of ANOVA with *p*-value < 0.05, conducted within each sampling. CON = Control; CHEM = chemical fertilizer; COMP = Compost.

**22nd June Sampling**				
**Treatment**	**Chlorophyll**	**Flavonols**	**Anthocyanins**	**NBI^®^**
CON	37.56 ± 2.1	1.22 ± 0.4	0.16 ± 0.01	42.81 ± 2.7
CHEM	37.00 ± 5.7	1.36 ± 0.2	0.16 ± 0.01	32.82 ± 5.5
COMP	38.22 ± 1.6	1.17 ± 0.6	0.16 ± 0.01	43.74 ± 3.2
**5th July Sampling**				
**Treatment**	**Chlorophyll**	**Flavonols**	**Anthocyanins**	**NBI^®^**
CON	37.24 ± 1.0	1.29 ± 0.1	0.18 ± 0.01	36.35 ± 1.6
CHEM	36.45 ± 4.1	1.29 ± 0.19	0.18 ± 0.01	36.35 ± 7.8
COMP	36.75 ± 2.3	1.28 ± 0.6	0.18 ± 0.003	37.99 ± 3.0

22nd June sampling Chlorophyll: F 0.08, df 8, P 0.92; Flavonols: F 1.74, df 8, P 0.25; Anthocyanins: F 0.33, df 8, P 0.72; NBI: F 2.1, df 8, P 0.2; 5th July sampling Chlorophyll: F 0.06, df 8, P 0.94; Flavonols: F 0.005, df 8, P 0.99; Anthocyanins: F 0.01, df 8, P 0.98 NBI: F 0.04, df 8, P 0.95.

**Table 3 insects-13-00717-t003:** Results of biomass collected and moisture content and relative standard deviation. In each column, values with different letters indicate mean values significantly different according to the Tukey’s post hoc test performed on results of ANOVA with *p*-value < 0.05. CON = Control; CHEM = chemical fertilizer; COMP = Compost.

Treatment	Stem (kg f.m.)	Moisture Content (%)	Inflorescences (kg f.m.)	Moisture Content (%)
CON	19.97 ± 1.76a	75.68 ± 4.73a	10.57 ± 1.02	78.95 ± 0.6
CHEM	24.57 ± 2.37b	76.83 ± 2.76a	13.07 ± 1.4	78.73 ± 2.13
COMP	24.97 ± 4.61b	79.12 ± 0.62b	12.7 ± 3.29	80.13 ± 1.83

Stem: F 5.38, df 8, P 0.04; Moisture content: F 8.1, df 8, P 0.019; Inflorescence: F 1.18, df 8, P 0.36 Moisture content: F 0.61, df 8, P 0.57.

**Table 4 insects-13-00717-t004:** Results of flower extracted nectar analysis. In each column, values with different letters indicate mean values significantly different according to the Tukey’s post hoc test performed on results of ANOVA with *p*-value < 0.05. Units are expressed as μg/flower, and the °Bx value is multiplied by 106. nd = not detectable. CON = Control; CHEM = chemical fertilizer; COMP = Compost.

	**Sampling 28 June**								
	**Oligosaccharides DP4 and DP5**	**Raffinose**	**Sucrose**	**Glucose**	**Fructose**	**Mannitol**	**Total Sugars**	**°Brix/Flower × 10** ** ^6^ **	**Total/°Bx**
**CON**	50 ± 25	25 ± 10	5 ± 2	270 ± 65	320 ± 130a	nd	670 ± 170a	850 ± 350	0.79 ± 0.19a
**CHEM**	25 ± 20	15 ± 10	5 ± 5	170 ± 100	230 ± 85a	nd	445 ± 230a	750 ± 200	0.59 ± 0.13b
**COMP**	30 ± 1	25 ± 4	10 ± 1	105 ± 95	175 ± 40b	nd	345 ± 5b	700 ± 200	0.49 ± 0.09b
	**Sampling 5 July**								
	**Oligosaccharides DP4 and DP5**	**Raffinose**	**Sucrose**	**Glucose**	**Fructose**	**Mannitol**	**Total Sugars**	**°Brix/Flower × 10^6^**	**Total/°Bx**
**CON**	25 ± 10	10 ± 5	5 ± 2	150 ± 35	270 ± 40a	2 ± 2	462 ± 85a	900 ± 100	0.51 ± 0.1
**CHEM**	30 ± 10	10 ± 5	5 ± 1	155 ± 2	250 ± 5a	2 ± 3	452 ± 20a	1050 ± 30	0.43 ± 0.01
**COMP**	35 ± 3	10 ± 3	5 ± 2	105 ± 30	180 ± 10b	2 ± 1	337 ± 28b	950 ± 60	0.35 ± 0.03

22nd June sampling Oligosaccharides: F 0.99, df 8, P 0.42; Raffinose: F 0.92, df 8, P 0.44; Sucrose: F 0.39, df 8, P 0.68; Glucose: F 4.1, df 8, P 0.07; Fructose: F 5.26, df 8, P 0.04; Total sugars: F 5.43, df 8, P 0.04; °Brix/flowers: F 0.27, df 8, P 0.76; Total/°Bx: F 5.17, df 8, P 0.04. 5th July sampling Oligosaccharides: F 0.65, df 8, P 0.55; Raffinose: F 0.40, df 8, P 0.68; Sucrose: F 0.58, df 8, P 0.58; Glucose: F 2.94, df 8, P 0.12; Fructose: F 11.9, df 8, P 0.008; Mannitol: F 0.07, df 8, P 0.93; Total sugars: F 6.03, df 8, P 0.03; °Brix/flowers: F 2.29, df 8, P 0.18; Total/°Bx: F 4.63, df 8, P 0.06.

**Table 5 insects-13-00717-t005:** Results of nectar analysis of the varietal study. Nd = not detectable. Units are expressed as μg/flower, the °Bx value is multiplied by 106.

	Sampling 28 June				Sampling 5 July			
Parameter	Hybrid Variety		Non-Hybrid Variety		Hybrid Variety		Non-Hybrid Variety	
	Mean	St.dev.	Mean	St.dev.	Mean	St.dev.	Mean	St.dev.
**Oligosaccharides DP4 and DP5**	50	40	25	15	105	15	85	10
**Raffinose**	30	25	10	10	20	4	20	10
**Sucrose**	12.5	5	7.5	5	10	1	10	4
**Glucose**	150	140	60	47	205	20	170	40
**Fructose**	190	110	115	51	390	80	265	30
**Mannitol**	Nd	Nd	40	48	10	2	5	1
**Total sugars**	433	320	258	174	740	121	555	95
**°Brix/flower × 10^6^**	700	350	550	100	1300	200	1100	100
**Total/°Bx**	0.62	0.23	0.47	0.04	0.57	0.02	0.50	0.01

## Data Availability

Data available on request from corresponding author.

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
