# Peer review of "Nectar Dynamics and Pollinators Preference in Sunflower"

_insects, 2022, doi:10.3390/insects13080717_

Round 1
Reviewer 1 Report
Thank you to the authors for their work.
If I understood well, you do not respond to your premise of studying the differences of varieties in the three components referred in the title (flower preference, nectar production and quality) in different soil conditions. Mainly, you analyzed the soil, biomass and nectar for a single variety and secondarily, separately analyzed nectar and flower visits between two varieties (hybrid and a non-hybrid, respectively). This experimental design does not respond to the questions you make. A suggestion would be to separate these two parts, but they would be very different than the one proposed here as one does not consider the varieties, and the other does not consider the soil changes.
Despite this vital issue that I have with the work, the flower preference is very neglected in relation to the other components. Assuming that you used all the space of the greenhouse for the “varietal study” (40.5m2), you have approximately 141 plants on each variety (half split) but you only observed 2 plants for variety every minute for 1.5 hours in 7 consecutive days. This is very limited. In the consequent results, I am missing the information on observed species or even functional groups (like, honeybees, bumblebees, other bees, syrphids, beetles, butterflies...). If analyzing the time variable here as you did, you should consider presenting the number of open flowers in the inflorescences every sampling moment (to get a sense of scale, for example, visits per 50 flowers), otherwise this should be compiled to a single value for the 7 days (again, referring a total number of flowers opened in the total time of the experiment).
Your approach on “pollinators” should be clarified; you are constantly considering exclusively honeybees throughout the text. There is no problem with that but you should clarify it, because there are many different nuances if you consider other species other than honeybees.
I am not a native english speaker, but particularly the “simple summary” and “abstract” need to be revised for the writing.
Some attention is needed to correct non-italic scientific names and double spaces between words, across all the text.
15
“climatic change, pesticides, pollutions, pest and diseases.” I would rewrite this to “climate change, inadequate pesticide use, invasive species and diseases”; pesticides might not be the reason per se, but its inadequate use; pollutions is very broad, I guess its already included in the first two reasons. Instead of “pest”, I would switch to “invasive species”, it is more present and relevant. “Pest” almost exclusively applied to honeybees in what concerns pollinators.
15
“The nectar produced by plants is the basic food for these insects, together with pollen”. This is not the case for all pollinators (for bees, yes, mainly pollen) such as syphids, most wasps, butterflies or beetles for example have other food sources during their lifetime.
20
“but also for the decline displayed globally in the last decades” can you specify what you are mentioning here? the honey production?
21
“soil condition” to “soil conditions”
31
“utilization” to “use”
40
“number of insects pollinating”, I would rather say “number of visits” as some insects might not be efficient pollinators.
44
“compost; pollinators; nectar composition; soil fertilization; sunflower”, please use other words other than the ones present in the title.
97
“growth ad floral”, I believe you wanted to say “and”
181
“The day” might be substituted by “At”
211-213
Which daily hour period did you cover in these 1.5 hours?
370-372 Fig1
I believe it is more correct to say “number of visits” instead of “number of insects”; you are counting the interactions, not being able to distinguish if the same insect visiting twice, for example. I suggest a reformulation of the figure. It does not need the tittle “pollinators visit”, the horizontal lines should all have the same aspect, the vertical axis should have some scale values on the side and both axis should have titles (suggestion here, “number of visits” and “date of sampling”).
Author Response
If I understood well, you do not respond to your premise of studying the differences of varieties in the three components referred in the title (flower preference, nectar production and quality) in different soil conditions. Mainly, you analyzed the soil, biomass and nectar for a single variety and secondarily, separately analyzed nectar and flower visits between two varieties (hybrid and a non-hybrid, respectively). This experimental design does not respond to the questions you make. A suggestion would be to separate these two parts, but they would be very different than the one proposed here as one does not consider the varieties, and the other does not consider the soil changes.
R: Thank you for the comment. The concept was not well explained and was misleading. Along the text the concept was clarified
Despite this vital issue that I have with the work, the flower preference is very neglected in relation to the other components. Assuming that you used all the space of the greenhouse for the “varietal study” (40.5m2), you have approximately 141 plants on each variety (half split) but you only observed 2 plants for variety every minute for 1.5 hours in 7 consecutive days. This is very limited. In the consequent results, I am missing the information on observed species or even functional groups (like, honeybees, bumblebees, other bees, syrphids, beetles, butterflies...). If analyzing the time variable here as you did, you should consider presenting the number of open flowers in the inflorescences every sampling moment (to get a sense of scale, for example, visits per 50 flowers), otherwise this should be compiled to a single value for the 7 days (again, referring a total number of flowers opened in the total time of the experiment).
R: Thank you for the comment. The study on the flower preference is indeed limited because it represents a preliminary study that we are improving for following seasons. To study the visit of insects we selected similar inflorescences in terms of diameter and size to be photographed. Only 7 days were considered because the number of visits was so decreases and the number of flowers open in the inflorescences selected too small that we decided to stop the recording. The number of insects is referred the sum of bumblebees and honeybees since we didn’t find any different species, anyway it wasn’t a study on groups of pollinators, so we didn’t focused on that. We understand the limit of the study and we are improving it for the next seasons considering colors, numbers of flowers open of the inflorescences, VOC emitted during recording; number and weight of seeds of the inflorescences at end of the crop cycle to better describe the pollination activity and the parameters that affect it. We believe that, even the limitation described, the study represents a preliminary approach to study pollinator preferences and to relate pollinators visits to nectar quality of different varieties, concept that is still not well described and clear. We clarified that along the text.
Your approach on “pollinators” should be clarified; you are constantly considering exclusively honeybees throughout the text. There is no problem with that but you should clarify it, because there are many different nuances if you consider other species other than honeybees.
R: Thank you for the comment. We referred to honeybees and bumblebees. We clarified that along the text.
I am not a native english speaker, but particularly the “simple summary” and “abstract” need to be revised for the writing.
R: Thank you for the comment the text was reworded
Some attention is needed to correct non-italic scientific names and double spaces between words, across all the text.
R: Thank you for the comment, the concept was clarified, and the text changed.
15
“climatic change, pesticides, pollutions, pest and diseases.” I would rewrite this to “climate change, inadequate pesticide use, invasive species and diseases”; pesticides might not be the reason per se, but its inadequate use; pollutions is very broad, I guess its already included in the first two reasons. Instead of “pest”, I would switch to “invasive species”, it is more present and relevant. “Pest” almost exclusively applied to honeybees in what concerns pollinators.
R: Thank you for the comment, the concept was clarified, and the text changed.
15
“The nectar produced by plants is the basic food for these insects, together with pollen”. This is not the case for all pollinators (for bees, yes, mainly pollen) such as syphids, most wasps, butterflies or beetles for example have other food sources during their lifetime.
R: Thank you for the comment, the concept was clarified, and the text changed.
20
“but also for the decline displayed globally in the last decades” can you specify what you are mentioning here? the honey production?
R: Thank you for the comment, the concept was clarified, and the text changed.
21
“soil condition” to “soil conditions”
R: Thank you for the comment, the concept was clarified, and the text changed.
31
“utilization” to “use”
R: Thank you for the comment, the concept was clarified, and the text changed.
40
“number of insects pollinating”, I would rather say “number of visits” as some insects might not be efficient pollinators.
R: Thank you for the comment, the concept was clarified, and the text changed.
44
“compost; pollinators; nectar composition; soil fertilization; sunflower”, please use other words other than the ones present in the title.
97
“growth ad floral”, I believe you wanted to say “and”
R: Thank you for the comment, the concept was clarified, and the text changed.
181
“The day” might be substituted by “At”
R: Thank you for the comment, the concept was clarified, and the text changed.
211-213
Which daily hour period did you cover in these 1.5 hours?
R: Thank you for the comment, the concept was clarified, and the text changed.
370-372 Fig1
I believe it is more correct to say “number of visits” instead of “number of insects”; you are counting the interactions, not being able to distinguish if the same insect visiting twice, for example. I suggest a reformulation of the figure. It does not need the tittle “pollinators visit”, the horizontal lines should all have the same aspect, the vertical axis should have some scale values on the side and both axis should have titles (suggestion here, “number of visits” and “date of sampling”).
R: Thank you for the comment, the graphic was not clear. The graphic was clarified and changed.
Reviewer 2 Report
The authors Bergonzoli et al. aims to evaluate the influence of soil characteristics and the genetic variety of Helianthus annus on nectar quality. My main concern with this manuscript ir that soil characteristics among treatments do not differ enough in order to evaluate the differences on nectar production, that indeed are small. May be an appropriate approach would have been to perform the greenhouse experiment that they set with both varieties in a random desing and including soils with different chemical characteristics. Considering that this might be difficult to perform I suggest to incorporate this drawbacks in the discussion section.
Some minnor suggestions are also stated:
-Along the ms please edit the species name, it should be in italics.
-Line 58 ...“ has ” decreased...
-Line 84 ...“ many” doubts...
-Line 97 ...“ and” floral density...
-Line 110 ...“ evaluating” ?
-Line 119. Revise this sentence "Sowing was performed by reaching a density of 7 plants?...”
-Line 181. please replace collected by measured.
Author Response
The authors Bergonzoli et al. aims to evaluate the influence of soil characteristics and the genetic variety of Helianthus annus on nectar quality. My main concern with this manuscript ir that soil characteristics among treatments do not differ enough in order to evaluate the differences on nectar production, that indeed are small. May be an appropriate approach would have been to perform the greenhouse experiment that they set with both varieties in a random desing and including soils with different chemical characteristics. Considering that this might be difficult to perform I suggest to incorporate this drawbacks in the discussion section.
Thanks for the comment, we better stressed at the end of the discussion section the intention to focus future studies on testing more varieties also with different soil treatments and evaluate pollinators behaviors on these treatments
Some minnor suggestions are also stated:
-Along the ms please edit the species name, it should be in italics.
Thanks for the comment. Done
-Line 58 ...“ has ” decreased...
-Line 84 ...“ many” doubts...
-Line 97 ...“ and” floral density...
-Line 110 ...“ evaluating” ?
-Line 119. Revise this sentence "Sowing was performed by reaching a density of 7 plants?...”
-Line 181. please replace collected by measured
Thanks for the comments, revisions were made
Reviewer 3 Report
This study presents several interesting results that will be of wide interest, notably that soil type was found to affect nectar composition, and the methodology is sound. While the data are robust, their presentation could be improved. The results require further clarification, and the discussion requires more extensive work to align with the aims and incorporate some text interpreting results that is currently presented in the results section. Additionally, the manuscript requires extensive editing of English language by an expert in the field to ensure that the messages conveyed are those intended by the author.
1. Introduction
Line 108 states that the study aims to better understand the soil-plant-pollinator microbial loop, whereas in fact only three of these four factors are investigated – soil, plant, and pollinator. The study does not contain a microbial experiment, and it is inappropriate to set this up as an aim of the study. Please write a clear aim/hypothesis for each of the two experiments that can be referred back to in the discussion.
2. Materials and methods
Line 115-122: Please provide some clarity around whether the soil within the plots was suitably homogenous, eg. will variation within the plot adversely affect the experiment?
Line 191 and 192 Please clarify whether the hybrid variety is the same as the one used in the soil experiments (907) and provide additional information on which hybrid and non-hybrid varieties were used.
2.9 data analysis – please specify which statistical tests were conducted for each individual experiment.
3. Results
Table 1 – here and throughout please include definitions of ‘CON’ ‘CHEM’ and ‘COMP’ in the legend. Make formatting consistent with tables 2 and 3. To make the significant differences more obvious, it could improve clarify to remove the ‘a’s for soil characters for which there was no significant difference between treatments (eg. clay, silt..)
Table 2 – please clarify whether the significance testing was conducted within the sample period, or across both of them. If it was across both of them, please also present data for within each sampling period.
Some interpretation of the results is present in the results section. This interpretive text should be moved into the discussion. Examples include lines 265-273, 281-282, and 356-358, however the entire section should be searched for additional examples of data interpretation.
Table 4 – please make the formatting consistent with tables 2 and 3. This change will involve swapping the row and column labels and merging the means and standard deviations into one cell so that the table columns are very similar to that of table 2.
Throughout the results section the results need to be more precisely described. For example, at lines 301 and 302 it is stated that there was a significant difference between treatments for total sugar concentration – please specify which treatments had higher/lower values and state means of the relevant treatments. This same approach needs to be taken throughout the results section – more details should be added and the text can be clarified.
Table 5 – Additional information about the significant differences is required. Does the first letter represent significant differences between treatments across the two sample periods? If two different significance tests are presented for one value, the use of different sets of symbols for each test may assist (eg. * for significance). Currently it is very confusing as ‘ab’ is usually interpreted as a treatment that is not significantly different to group a or group b.
Lines 352-353 state that in the varietal experiment there were no significant differences found between treatments in the nectar analysis. This conflicts with line 356 which talks about differences in nectar composition. Perhaps line 356 is in reference to the other experiment, in which case this should be clarified.
Line 363 mentions a ‘rustic variety’, however in the methods a non-hybrid variety is mentioned (line 192), and in the graph the term ‘old population’ is used. Assuming these references are to the same thing, please keep terminology consistent throughout the manuscript.
Line 365 and 366 mention a significant difference in pollinator numbers. Please provide details of what statistical test was conducted in the methods, and present the results of these tests including P values in the results.
4. Discussion
The discussion shows promise and presents exciting results, however requires extensive revision of structure and content prior to publication. The discussion should clearly address the aims/hypotheses that will be set up in the introduction (see comments on introduction). The key conclusions need to be clearly presented – currently they are buried in the manuscript. For example lines 394-396 contain a key result that soil had a significant effect on nectar composition, which could be much more clearly presented upfront. As per comments in the results section, interpretation of results needs to be integrated into the discussion. While the discussion requires a re-write, some specific comments below should be helpful during this process:
Line 408 refers to ‘these findings’ to avoid confusion and provide clarity, here and throughout the manuscript always refrain from using ‘this’ and ‘these’ – specify exactly which result you are referring to.
Line 419-420 discuss other factors that may influence pollinator visitation. Another factor that has not been considered is chemical cues that were not detected in the nectar analysis, which is designed to only detect a subset of compounds. Please include this factor.
Line 397 and 422 reference a ‘microbial loop’. As per comments on the introduction please ensure this is used appropriately. If this loop is to be discussed include an explanation of what it is, what the results of the present study suggest, and why, and integrate this with the current literature.
Abstract
The two specific aims should be clarified as per comments on the introduction and discussion.
Short summary
Line 22 refers to a positive effect of compost on pollinator preference. This study did not test for a relationship between compost, nor flowers grown under the compost treatment, and pollinator preference. Please update the text. If this is an inferred relationship, this should be stated, and adequate supporting text provided in the discussion.
Author Response
This study presents several interesting results that will be of wide interest, notably that soil type was found to affect nectar composition, and the methodology is sound. While the data are robust, their presentation could be improved. The results require further clarification, and the discussion requires more extensive work to align with the aims and incorporate some text interpreting results that is currently presented in the results section. Additionally, the manuscript requires extensive editing of English language by an expert in the field to ensure that the messages conveyed are those intended by the author.
- Introduction
Line 108 states that the study aims to better understand the soil-plant-pollinator microbial loop, whereas in fact only three of these four factors are investigated – soil, plant, and pollinator. The study does not contain a microbial experiment, and it is inappropriate to set this up as an aim of the study. Please write a clear aim/hypothesis for each of the two experiments that can be referred back to in the discussion.
R: thank you for the comment. The goal of the study was clarified.
- Materials and methods
Line 115-122: Please provide some clarity around whether the soil within the plots was suitably homogenous, eg. will variation within the plot adversely affect the experiment?
R: Thank you for the comment. The concept was clarified also in result section
Line 191 and 192 Please clarify whether the hybrid variety is the same as the one used in the soil experiments (907) and provide additional information on which hybrid and non-hybrid varieties were used.
R: Thank you for the comment, the concept was clarified and better described.
2.9 data analysis – please specify which statistical tests were conducted for each individual experiment.
R: Thank you for the comment, the concept was clarified and better described.
- Results
Table 1 – here and throughout please include definitions of ‘CON’ ‘CHEM’ and ‘COMP’ in the legend. Make formatting consistent with tables 2 and 3. To make the significant differences more obvious, it could improve clarify to remove the ‘a’s for soil characters for which there was no significant difference between treatments (eg. clay, silt..
)
Thanks for the comment, legend was included in all tables, style and formatting have been adjusted and letter “a” was removed where soil non-significant differences among treatments were displayed.
Table 2 – please clarify whether the significance testing was conducted within the sample period, or across both of them. If it was across both of them, please also present data for within each sampling period.
R: Thank you for the comment, the concept was clarified and better described. The analysis was not performed across the sample periods because the difference would be effect of the phenological phase and not to of the treatments.
Some interpretation of the results is present in the results section. This interpretive text should be moved into the discussion. Examples include lines 265-273, 281-282, and 356-358, however the entire section should be searched for additional examples of data interpretation.
Thanks for the comments. These parts were moved to discussion and additional examples for data interpretations were provided
Table 4 – please make the formatting consistent with tables 2 and 3. This change will involve swapping the row and column labels and merging the means and standard deviations into one cell so that the table columns are very similar to that of table 2.
Thank you for the comments, changes were applied
Throughout the results section the results need to be more precisely described. For example, at lines 301 and 302 it is stated that there was a significant difference between treatments for total sugar concentration – please specify which treatments had higher/lower values and state means of the relevant treatments. This same approach needs to be taken throughout the results section – more details should be added and the text can be clarified.
R: Thank you for the comment, the concept was clarified and better described
Table 5 – Additional information about the significant differences is required. Does the first letter represent significant differences between treatments across the two sample periods? If two different significance tests are presented for one value, the use of different sets of symbols for each test may assist (eg. * for significance). Currently it is very confusing as ‘ab’ is usually interpreted as a treatment that is not significantly different to group a or group b.
R: Thank you for the comment, the concept was clarified and better described along the text and in the tables 4 and 5
Lines 352-353 state that in the varietal experiment there were no significant differences found between treatments in the nectar analysis. This conflicts with line 356 which talks about differences in nectar composition. Perhaps line 356 is in reference to the other experiment, in which case this should be clarified.
R: Thank you for the comment, the concept was clarified and better described.
Line 363 mentions a ‘rustic variety’, however in the methods a non-hybrid variety is mentioned (line 192), and in the graph the term ‘old population’ is used. Assuming these references are to the same thing, please keep terminology consistent throughout the manuscript.
R: Thank you for the comment, the concept was clarified, and the text changed.
Line 365 and 366 mention a significant difference in pollinator numbers. Please provide details of what statistical test was conducted in the methods, and present the results of these tests including P values in the results.
R: Thank you for the comment, the concept was clarified, and the text changed.
- Discussion
The discussion shows promise and presents exciting results, however requires extensive revision of structure and content prior to publication. The discussion should clearly address the aims/hypotheses that will be set up in the introduction (see comments on introduction). The key conclusions need to be clearly presented – currently they are buried in the manuscript. For example lines 394-396 contain a key result that soil had a significant effect on nectar composition, which could be much more clearly presented upfront. As per comments in the results section, interpretation of results needs to be integrated into the discussion. While the discussion requires a re-write, some specific comments below should be helpful during this process:
Line 408 refers to ‘these findings’ to avoid confusion and provide clarity, here and throughout the manuscript always refrain from using ‘this’ and ‘these’ – specify exactly which result you are referring to.
Line 419-420 discuss other factors that may influence pollinator visitation. Another factor that has not been considered is chemical cues that were not detected in the nectar analysis, which is designed to only detect a subset of compounds. Please include this factor.
Line 397 and 422 reference a ‘microbial loop’. As per comments on the introduction please ensure this is used appropriately. If this loop is to be discussed include an explanation of what it is, what the results of the present study suggest, and why, and integrate this with the current literature.
Thanks for the comments, the discussion were totally re-written according to your indications. Specifically, we tried to discuss step by step the results obtained giving relevance to the main findings but without neglecting some missing elements on which we are working on.
Abstract
The two specific aims should be clarified as per comments on the introduction and discussion.
Thank you, the two aims were better specified
Short summary
Line 22 refers to a positive effect of compost on pollinator preference. This study did not test for a relationship between compost, nor flowers grown under the compost treatment, and pollinator preference. Please update the text. If this is an inferred relationship, this should be stated, and adequate supporting text provided in the discussion.
Thank you for the comment, the short summary was clarified
Round 2
Reviewer 1 Report
Thank you to the authors for the work put into this work.
Despite all the efforts that authors made to respond to the several reviewers, I believe the manuscript is not consistent and the experimental design is very limited in what concerns the “insect” part. The n for this part is too low and not enough to sustain such results. Mainly for that reason, I believe this work does not fit in the journal “Insects”.
23
I believe that a more realistic reason related with the pollinators might be, not the honey, but the insufficient quantity and quality of food resources (nectar and pollen) due to landscape changes, particularly large-scale sunflower explorations. I believe it adds more credibility to your work, by you responding to some of the dynamics.
208
I believe you need to refer the pollinators here, otherwise you are making a methodology not compatible with just a nectar composition study.
226
You should point the taxonomic names (Apis mellifera and Bombus spp.).
409
You changed the name of this section before.
450
Honeybee is only one species in this location, Bumblebees might be several species (unless you identified them). You should refer to them not as two species but as two different pollinators groups.
Author Response
Reviewer 1
Thank you to the authors for the work put into this work.
Despite all the efforts that authors made to respond to the several reviewers, I believe the manuscript is not consistent and the experimental design is very limited in what concerns the “insect” part. The n for this part is too low and not enough to sustain such results. Mainly for that reason, I believe this work does not fit in the journal “Insects”.
23 I believe that a more realistic reason related with the pollinators might be, not the honey, but the insufficient quantity and quality of food resources (nectar and pollen) due to landscape changes, particularly large-scale sunflower explorations. I believe it adds more credibility to your work, by you responding to some of the dynamics.
-Thank you, the words pollen and nectar were added
208 I believe you need to refer the pollinators here, otherwise you are making a methodology not compatible with just a nectar composition study.
-Thank you, the reference to pollinators was re-included
226 You should point the taxonomic names (Apis mellifera and Bombus spp.).
-Thank you, done
409 You changed the name of this section before.
-Thank you, changed
450 Honeybee is only one species in this location, Bumblebees might be several species (unless you identified them). You should refer to them not as two species but as two different pollinators groups
-Thank you, a change was applied
Reviewer 2 Report
The authors really improved their manuscript and have sufficiently addressed my comments. I have some minor issues that you can find below. I would also suggest the authors to revise English grammar along the manuscript in order to facilitate reading.
Line 124 The sentence could be joined with the previous paragraph.
Line 222 Change "in the other.." by "in the field experiment".
Line 415. Please check if between "as" and "g/flower" there is a double space.
Line 485. Change "determined" by "evidenced".
Line 489. Should say "indicators".
Line 496. Please eliminate "..." should say ", among others"
Line 523. Should say "..even in a very low..".
Line 527. Should say "pollinator".
Line 528. Should say "with respect to a".
Lines 537-538. Better "...to favor the production of even more..."
Author Response
Reviewer 2
The authors really improved their manuscript and have sufficiently addressed my comments. I have some minor issues that you can find below. I would also suggest the authors to revise English grammar along the manuscript in order to facilitate reading.
Line 124 The sentence could be joined with the previous paragraph.
-Thank you, done
Line 222 Change "in the other.." by "in the field experiment".
-Thank you, the sentence was delated and revised in revision 1 according to another reviewer request
Line 415. Please check if between "as" and "g/flower" there is a double space.
-Thank you, verified
Line 485. Change "determined" by "evidenced".
-Thank you, done
Line 489. Should say "indicators".
-Thank you, the sentence now refers to C and N as indicators
Line 496. Please eliminate "..." should say ", among others"
-Thank you, done
Line 523. Should say "..even in a very low..".
-Thank you, done
Line 527. Should say "pollinator".
-Thank you, done
Line 528. Should say "with respect to a".
-Thank you, done
Lines 537-538. Better "...to favor the production of even more..."
-Thank you, done
Reviewer 3 Report
The authors have implemented clear major improvements to the manuscript, and it is heading in the right direction. Further clarification of the results and discussion is required, as is further development of the aims – each experiment that was conducted needs to be clearly presented and consistently referenced throughout the manuscript.
Lines 114-125: While they are improved, further work is required on the aims. There is currently no mention of testing nectar composition of sunflowers grown on different soil treatments. This section needs to be clear and not repetitious. It would be helpful to set up clear names for each experiment either here or elsewhere in the manuscript so they can be easily referenced later. I query the necessity of lines 124-125. In line 116, what specific relationship? It is best general practice to avoid ambiguous references to concepts in other sentences, particularly in an aims section. I do not believe it is appropriate to set up honeybees and bumblebees as an aim – this is the result. You looked at pollinators, and the result was that you only detected bees, you were not aiming to only detect bees. Also see my later comment about the appropriateness of the ‘pollinator study’, it would be good to clarify the intended aims upfront early on.
Lines 193:194: ‘Sampling of flowers was performed two times, on 28th June and on 5th July.’ This information belongs in the first paragraph about sampling, not in the latter about HPLC and SSR.
2.5 Biomass collection: it is unclear what was done here. I note the change from measured to collected. Were the plants harvested? This would be a more appropriate term if so. How were the measurements conducted? How were the leaves dealt with? What scales/other equipment were used?
2.7 Pollinator study. This section should be renamed and restructured. This is actually two separate experiments that should be dealt with independently. The first (and most robust) experiment is about nectar composition of hybrid and non-hybrid plants, and does not relate to pollinators – it is misleading to imply that it does. The second experiment (more limited in scope) is about pollinator visitation to hybrid and non-hybrid plants. These two experiments should be presented separately to avoid confusion.
Line 224-225, specify which other experiment by referencing the relevant section number. A total of six what?
Line 226: remove reference to bees, as previously flagged this is a result not an aim or method. Additionally, please specify exactly how many flowers of each treatment were monitored during this study.
Line 235, what was the analysis methodology? Please provide details of how the data were collected.
2.9 data analyses: specify which experiments you are talking about (can include reference to sections as appropriate). It is unclear which data are being referenced.
Results:
Throughout this section, when you mention significant results please provide the associated test statistic, p-value, and mean. When making reference to higher or lower values, please state what they are.
Lines 259-260: ‘All the treatments shown an increase of each parameter studied respect to the sampling before ploughing’ confusing grammatical errors aside (please get the manuscript checked by a native speaker), this statement appears to be false. Higher clay values were returned before ploughing than after for CHEM and COMP, with similar inconsistencies in the silt, sand, and pH factors for some treatments.
Lines 260-261, 268-270, 274-275: do not provide interpretation of results, save this for the discussion.
Line 313: assume this means no significant differences?
Table 4: as you did in the first table, please remove additional ‘a’s for treatments in which no significant difference was found.
Lines: 323-326: this is a good example of specifying the means in the results, please ensure this approach is consistently applied throughout the results section. However, p-values still require adding. It is also unclear whether CON was also significantly different.
Lines 431-432: this is the key result, put it first at the beginning of the section. Also clarify what treatments you mean – you can say between the hybrid and non-hybrid variety and between the two sampling periods.
Line 449: How many visits were detected in total? How many by each pollinator? How many hours of observations were conducted in total?
Figure 1: why are there 3 versions of this figure? Y axis can be updated – no of pollinator visits
Lines 476: I would rephrase ‘evaluate pollinator preferences’. The nectar comparison of hybrid/non hybrid presented very interesting results. Focus on these. The pollinator experiment was much smaller in scope and the inferences that can be drawn from it are more limited.
Lines 506: It is unclear which nectar characterisation you are referring to. As suggested above, develop clear names for each experiment and use them consistently throughout. You need to differentiate between the experiment testing the effect of soil treatment on nectar composition, and the one testing the effect of hybrid/non-hybrid on nectar composition.
Line 511: similarly, what is the main analysis?
Line 506: please rewrite this paragraph to clarify the results and their implications and think carefully about each sentence and if it makes sense. I am not sure what the paragraph is supposed to be about. The first line references fertilization (hybrid vs no hybrid?) and the rest of the text discusses soil treatments. There are some very interesting results that are not clearly presented – eg that comp decreased sugars (in both sampling periods). Clearly state the main results upfront. Then discuss intricacies (eg. differences between sampling periods). What are the implications? The meaning of the last sentence ‘This is a further factor that highlight the effect of soil quality and its fertilization strategy on nectar composition’ is not clear. The fertilisation strategy does not belong to soil quality.
Line 527: this is actually two studies – one about nectar composition, and one about pollinator visitation. Present and discuss them separately to avoid misleading readers. You need to summarise the results more clearly throughout the discussion. Eg. instead of saying the varieties, specify the non hybrid and hybrid variety. Say there was no difference in the analysed nectar traits between the non hybrid and hybrid variety. Be clear and specify what you are talking about.
Line 530: ‘intended as’ does not make sense. Was this an intent of the study? I don’t believe so. The word is inappropriate. Please get the entire manuscript edited by a native speaker with knowledge of the field. Without doing so misinterpretations and overstatements will continue to occur.
It would be appropriate to flag the small sample size of the pollinator observation study as a limitation.
Line 41-42: ‘fertilization strategy influenced crop development, soil quality and nectar composition’ I don’t believe these relationships were measured? Soil quality and crop development (uncertain which specific results this pertains to) were not tested with hybrid/non-hybrid varieties?
Line 43: ‘very little differences’ can you say no significant differences
Line;44-45, climate and agricultural management (other than soil which is already mentioned) were not tested in the present study.
Lines 42-45: despite the above-mentioned deficiencies, this sentence clearly states and interprets a result of the study. It would be good to see this conclusion clearly presented and discussed in the discussion.
Author Response
Lines 114-125: While they are improved, further work is required on the aims. There is currently no mention of testing nectar composition of sunflowers grown on different soil treatments. This section needs to be clear and not repetitious. It would be helpful to set up clear names for each experiment either here or elsewhere in the manuscript so they can be easily referenced later. I query the necessity of lines 124-125. In line 116, what specific relationship? It is best general practice to avoid ambiguous references to concepts in other sentences, particularly in an aims section. I do not believe it is appropriate to set up honeybees and bumblebees as an aim – this is the result. You looked at pollinators, and the result was that you only detected bees, you were not aiming to only detect bees. Also see my later comment about the appropriateness of the ‘pollinator study’, it would be good to clarify the intended aims upfront early on.
-Thank you. The testing of nectar composition of sunflower grown in different treated soils was mentioned. The end of the introduction has been re-organized to separate each purpose and each study performed in order to clarify the work done. Your observation is correct to cite only pollinator, we previously updated according to another review request, but we believe that is more correct to keep the word pollinator
Lines 193:194: ‘Sampling of flowers was performed two times, on 28th June and on 5th July.’ This information belongs in the first paragraph about sampling, not in the latter about HPLC and SSR.
-Thank you, done
2.5 Biomass collection: it is unclear what was done here. I note the change from measured to collected. Were the plants harvested? This would be a more appropriate term if so. How were the measurements conducted? How were the leaves dealt with? What scales/other equipment were used?
-Thank you, the information required were added
2.7 Pollinator study. This section should be renamed and restructured. This is actually two separate experiments that should be dealt with independently. The first (and most robust) experiment is about nectar composition of hybrid and non-hybrid plants, and does not relate to pollinators – it is misleading to imply that it does. The second experiment (more limited in scope) is about pollinator visitation to hybrid and non-hybrid plants. These two experiments should be presented separately to avoid confusion.
-Thank you, the section was modified, and a further subsection was added. The two experiments are now described separately.
Line 224-225, specify which other experiment by referencing the relevant section number. A total of six what?
-Thank you, the sentence was modified and the concept clarified
Line 226: remove reference to bees, as previously flagged this is a result not an aim or method. Additionally, please specify exactly how many flowers of each treatment were monitored during this study.
-Thank you, done
Line 235, what was the analysis methodology? Please provide details of how the data were collected.
-Thank you, corrected
2.9 data analyses: specify which experiments you are talking about (can include reference to sections as appropriate). It is unclear which data are being referenced.
-Thank you, corrected
Results:
Throughout this section, when you mention significant results please provide the associated test statistic, p-value, and mean. When making reference to higher or lower values, please state what they are.
-Thank you, corrected
Lines 259-260: ‘All the treatments shown an increase of each parameter studied respect to the sampling before ploughing’ confusing grammatical errors aside (please get the manuscript checked by a native speaker), this statement appears to be false. Higher clay values were returned before ploughing than after for CHEM and COMP, with similar inconsistencies in the silt, sand, and pH factors for some treatments.
-Thank you, the sentence was reworded
Lines 260-261, 268-270, 274-275: do not provide interpretation of results, save this for the discussion.
-Thank you, corrected
Line 313: assume this means no significant differences?
-Thank you, yes, the word significant was added
Table 4: as you did in the first table, please remove additional ‘a’s for treatments in which no significant difference was found.
-Thank you, done
Lines: 323-326: this is a good example of specifying the means in the results, please ensure this approach is consistently applied throughout the results section. However, p-values still require adding. It is also unclear whether CON was also significantly different.
-Thank you, reference of means values have been now used through the text to present the results. The p-value was specified in the material and method section. The sentence related to CON treatment was clarified
Lines 431-432: this is the key result, put it first at the beginning of the section. Also clarify what treatments you mean – you can say between the hybrid and non-hybrid variety and between the two sampling periods.
-Thank you, there is a problem in lines offset due to editing and is not easy to identify the sentence. In any case, if the sentence refers to:
Regarding the results of nectar analysis of the varietal study, no statistically significant differences were found between treatments. Comparison among results of the two samplings shown an increase of the values of Glucose, Fructose, Total sugars and Brix for each treatment in the second sampling with statistically significant differences (p < 0.05).
It was modified as follows:
Regarding the results of nectar analysis in hybrid and non-hybrid varieties, the comparison among the two samplings periods shown an increase of the values of Glucose, Fructose, Total sugars and Brix for each treatment in the second sampling with statistically significant differences. On the other hand, no significant differences were found between treatments
Line 449: How many visits were detected in total? How many by each pollinator? How many hours of observations were conducted in total?
-Thank you, the information was given in the text
Figure 1: why are there 3 versions of this figure? Y axis can be updated – no of pollinator visits
-Thank you, done
Lines 476: I would rephrase ‘evaluate pollinator preferences’. The nectar comparison of hybrid/non hybrid presented very interesting results. Focus on these. The pollinator experiment was much smaller in scope and the inferences that can be drawn from it are more limited.
-Thank you, the sentence was redesigned
Lines 506: It is unclear which nectar characterisation you are referring to. As suggested above, develop clear names for each experiment and use them consistently throughout. You need to differentiate between the experiment testing the effect of soil treatment on nectar composition, and the one testing the effect of hybrid/non-hybrid on nectar composition.
-Thank you, a refence to table 4 has been included
Line 511: similarly, what is the main analysis?
-Thank you, also in this case we referred to the table 4
Line 506: please rewrite this paragraph to clarify the results and their implications and think carefully about each sentence and if it makes sense. I am not sure what the paragraph is supposed to be about. The first line references fertilization (hybrid vs no hybrid?) and the rest of the text discusses soil treatments. There are some very interesting results that are not clearly presented – eg that comp decreased sugars (in both sampling periods). Clearly state the main results upfront. Then discuss intricacies (eg. differences between sampling periods). What are the implications? The meaning of the last sentence ‘This is a further factor that highlight the effect of soil quality and its fertilization strategy on nectar composition’ is not clear. The fertilisation strategy does not belong to soil quality.
-Thanks for the comment, the paragraph refers only to the interpretation of table 3 and 4, which take into account only the study of the hybrid variety. The sentence has been written newly
Line 527: this is actually two studies – one about nectar composition, and one about pollinator visitation. Present and discuss them separately to avoid misleading readers. You need to summarise the results more clearly throughout the discussion. Eg. instead of saying the varieties, specify the non hybrid and hybrid variety. Say there was no difference in the analysed nectar traits between the non hybrid and hybrid variety. Be clear and specify what you are talking about.
-Thanks for the comment, even if we believe that nectar study (in hybrid and non-hybrid variety) and pollinator visits are strictly connected, we tried to separate the two findings discussing first the results of nectar composition and then the pollinator visit. The main aspect we wanted to highlight was that despite similar nectar composition, much more insects visited the non-hybrid variety. This difference is therefore attributed not to the variety itself, but to other factors such as number of flowers, colors of flower, VOC emitted, flower morphology. We will deeply study these aspects in future research.
Line 530: ‘intended as’ does not make sense. Was this an intent of the study? I don’t believe so. The word is inappropriate. Please get the entire manuscript edited by a native speaker with knowledge of the field. Without doing so misinterpretations and overstatements will continue to occur.
-Thank you, revised
It would be appropriate to flag the small sample size of the pollinator observation study as a limitation.
-Thank you, included
Line 41-42: ‘fertilization strategy influenced crop development, soil quality and nectar composition’ I don’t believe these relationships were measured? Soil quality and crop development (uncertain which specific results this pertains to) were not tested with hybrid/non-hybrid varieties?
-Thank you, adjusted
Line 43: ‘very little differences’ can you say no significant differences
-Thank you, done
Line;44-45, climate and agricultural management (other than soil which is already mentioned) were not tested in the present study.
-Thank you, the sentence was delated
Lines 42-45: despite the above-mentioned deficiencies, this sentence clearly states and interprets a result of the study. It would be good to see this conclusion clearly presented and discussed in the discussion.
-Thank you, discussion includes now this explanation